# An image segmentation technique with statistical strategies for pesticide efficacy assessment

**Steven B. Kim[1], Dong Sub Kim [2]\*, Xiaoming Mo[1]**

**1** Department of Mathematics and Statistics, California State University, Monterey Bay, Seaside, California, United States of America, **2** Department of Plant Sciences, University of California, Davis, Salinas, California, United States of America

\* vdskim@ucdavis.edu

**Data Availability Statement:** All relevant data are within the manuscript and its Supporting information files.

**Funding:** The author(s) received no specific funding for this work.

## Abstract

Image analysis is a useful technique to evaluate the efficacy of a treatment for weed control. In this study, we address two practical challenges in the image analysis. First, it is challenging to accurately quantify the efficacy of a treatment when an entire experimental unit is not affected by the treatment. Second, RGB codes, which can be used to identify weed growth in the image analysis, may not be stable due to various surrounding factors, human errors, and unknown reasons. To address the former challenge, the technique of image segmentation is considered. To address the latter challenge, the proportion of weed area is adjusted under a beta regression model. The beta regression is a useful statistical method when the outcome variable (proportion) ranges between zero and one. In this study, we attempt to accurately evaluate the efficacy of a 35% hydrogen peroxide (HP). The image segmentation was applied to separate two zones, where the HP was directly applied (gray zone) and its surroundings (nongray zone). The weed growth was monitored for five days after the treatment, and the beta regression was implemented to compare the weed growth between the gray zone and the control group and between the nongray zone and the control group. The estimated treatment effect was substantially different after the implementation of image segmentation and the adjustment of green area.

## Introduction

Ryegrass (*Lolium multiflorum*) is one of the most predominant weeds in United States [1] and pesticides are used to control ryegrass. Pesticide efficacy assessments are important to measure the efficacy of a treatment for ryegrass control. Pesticide efficacy have been usually determined by manual weed counting at a given time since the application of a treatment. Manual counting, however, is labor intensive and time-consuming, and an alternative method of data collection is images (i.e., taking a picture). Image analysis has been applied for the discrimination between crops and weeds [2], weed detection [3–6], weed mapping [7, 8], and identification of weed species and patches [9] in agricultural fields. Technologies of image analysis make data

**Competing interests:** The authors have declared that no competing interests exist.

collection faster and easier than the manual counting, and the image analysis can provide more information about the efficacy of a treatment particularly when the number of experimental units is limited.

If the pesticide does not affect an entire experimental unit, and if the researcher assesses the efficacy on the entire experimental unit, the measure of efficacy can be inaccurate and/or unstable. In this regard, the image analysis will be more useful than manual counting. For example, a pesticide application through drip irrigation in commercial agricultural systems can show spotty efficacy. The pesticide would not be much different than the control if the weed density was assessed by manual counting. Two distinctive zones may appear within an experimental unit, where the weed is controlled near the hole of the drip tapes and where the weed is abundant away from the hole. In order to accurately estimate the efficacy of the pesticide, the two zones need to be separated in the image analysis. Otherwise, the efficacy of the pesticide can be severely underestimated. The two zones can be separated by image segmentation techniques which organize pixels of an image to several categories according to color, brightness, and texture [10]. Onyango and Marchant [11] demonstrated that green vegetation and soil can be distinguished by image segmentation. Several approaches using deep neural networks have proven to be effective for segmentation task [12–14]. Moreover, detection of ryegrasses in crops using machine learning technology [15] and detection of other weeds in ryegrasses using deep learning technology [16] have been reported. However, upon our best knowledge, the image segmentation technique has not been used with a statistical method which adjusts a biased estimate of green cover at early days of application for the pesticide efficacy assessment. pesticide efficacy assessment. If the image segmentation is applied for the examples, an error that the pesticide is not effective for weed control will not be given. Furthermore, multiple parameters can be considered to optimally utilize a fixed amount of resources (e.g., frequency, concentration, irrigation methods (surface, buried, or spray), and the number of drip tapes).

In addition to the challenge due to the spotty efficacy, RGB codes, which can be used to identify weed growth in the image analysis, may not be stable due to various surrounding factors, human errors, and unknown reasons. The colors of objects can vary depending on energy sources such as sun light and artificial lamps, light intensity, color balance, direction, and more [17]. In other words, there is an unexpected measurement error in the quantification of weed growth in addition to the natural variability in the weed growth.

There has been an arbitrary decision on the time of pesticide efficacy assessment on suppression of weed seed germination. If the assessment is done too early, it would be difficult to compare the treatment and the control because the weed has not grown yet. The treatment would not last forever, and the effect size relative to the control may be a function of time. Therefore, it would be critical to monitor the treatment effect with respect to time (i.e., longitudinal study) instead of choosing an arbitrary time point for pesticide efficacy assessment (i.e., cross-sectional study). In this regard, the image analysis becomes useful because taking a picture multiple times is relatively simple for longitudinal study (less laborious than the manual counting). A longitudinal study allows researchers to estimate the rate of weed growth which cannot be achieved by a cross-sectional study.

If the image analysis is applied to pesticide efficacy assessment, one parameter of interest can be the proportion of area occupied by weeds per experimental unit. Typically, soil color (brown) and weed color (green) are clearly distinguishable, so the RGB codes can be used to estimate the proportion of green area [18]. The outcome variable is a proportion (e.g., a fraction of green color pixels out of the total pixels in an image) which is bounded between zero and one. Beta regression is a special kind of regression which can model data especially when their values are between zero and one [19]. When the traditional linear regression is used in

this case, it can lead to biased estimation for the expected outcome. The beta regression can model the expected proportion of green area (possible values are between zero and one) with respect to days after treatment. However, due to potential background noise in the measurement of RGB codes or other factors which make a portion of image appears green, it is possible to observe a nonzero proportion of green area on day zero of an experiment. In this regard, we attempt to re-parameterize the model to adjust the expected proportion of green area in order to accurately compare the treatment and the control.

The purpose of this article is to demonstrate the image analysis (monitoring pictures of weeds over time, implementing image segmentation, and extracting RGB codes) and statistical modeling (beta regression and adjusting the expected proportion of green area) in the pesticide efficacy assessment. For the purpose of demonstration, we evaluated the efficacy of hydrogen peroxide (HP) on suppressing the weed growth. Despite small sample sizes, four replicates for the treatment and four replicates for the control, we could conclude that the effect of HP is statistically significant based on the image analysis and statistical modeling.

## Materials and methods

### Treatment description and data collection

Ryegrass (*Lolium multiflorum*) seeds were sowed on pots (7.5 × 7.5 cm). 12 mL of 35% hydrogen peroxide (HP) was applied by 50 mL syringe at the center of each pot on September 3, 2020. The experimental design was a randomized complete block with four replications (pots) for the control and four replications for the HP treatment. Each of the four pots was photographed by a digital camera (EOS 70D DSLR, Cannon, Inc., Tokyo, Japan) one, two, four, and five days after treatment (DAT) (Figs 1 and 2) under the lights of four incandescent bulbs of 20 watts. Each GIF file, converted from a JPEG file, was uploaded to the image analysis program available at http://mkwak.org/imgarea/. The program outputs RGB codes (which represent various colors quantitatively as a combination of red, green and blue) and the number of pixels associated with each RGB code.

### Image segmentation

In the flowerpots where the HP treatment was applied, there were two clearly distinguishable soil colors, gray and nongray (mostly brown), due to the decolorization induced by hydroxyl radicals [20]. The two zones (gray zone and nongray zone) were separated as shown in Figs 1 and 2, and this image segmentation could be implemented easily by the 'Remove Background' tool in Microsoft PowerPoint. The process is automatic and has been tried several times to obtain the best segmentation.

### Statistical analysis

In each flowerpot treated by the HP, the gray zone (referred to as $P_1$) well absorbed the HP treatment, and the nongray zone (referred to as $P_0$) did not well absorb the treatment. The two zones were clearly distinguishable by soil color. The objective of data analysis was to compare the weed growth among $P_1$, $P_0$, and control (referred to as C).

The outcome variable of interest is the observed proportion of green area per image. A "green" color was identified by the six most common RGB codes such that G > R and G > B. The observed proportion of green area was calculated by dividing the total "green" pixels by the total pixels per image, and this outcome variable ranges between zero and one.

To respect the range of the outcome variable, between zero and one, it may not be appropriate to use linear regression because an estimated proportion can be below zero or above one

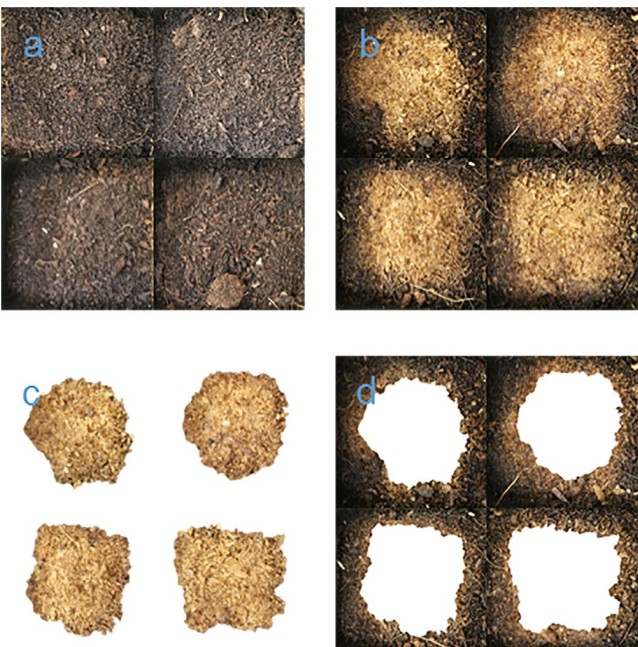

**Fig 1. The pictures of 1 DAT.** a: Control; b: HP treatment (before segmentation); c: Area directly affected by HP (P1; after segmentation); d: Area not directly affected by HP (P0; after segmentation).

under the linearity assumption (see Fig 3). Instead, beta regression can be more appropriate for the outcome variable which ranges between zero and one [16]. When the beta regression is applied to the observed data, the outcome variable (to be predicted) is the observed proportion of green area per image, and the predictor variables are days after treatment and treatment group (C, $P_0$, or $P_1$). The expected proportion of green area is denoted by $\mu$, and $h(\mu) = \ln[\mu / (1 - \mu)]$ is modeled by the linear predictor $\alpha_0 + \alpha_1 P_0 + \alpha_2 P_1 + \beta_0 D + \beta_1 (D \times P_0) + \beta_2 (D \times P_1)$ in the beta regression. The regression parameters $(\alpha_0, \alpha_1, \alpha_2, \beta_0, \beta_1, \beta_2)$ are to be estimated given data, D is a continuous numeric variable which represents days after treatment, $P_0$ is a dummy variable such that $P_0 = 1$ for the nongray zone of the HP treatment ($P_0 = 0$ otherwise), and $P_1$ is another dummy variable such that $P_1 = 1$ for the gray zone of the HP treatment ($P_1 = 0$ otherwise). The linear predictor represents the three beta regression models: $h(\mu) = \alpha_0 + \beta_0 D$ for the control group ($P_0 = 0$; $P_1 = 0$), $h(\mu) = (\alpha_0 + \alpha_1) + (\beta_0 + \beta_1) D$ for the nongray zone of HP ($P_0 = 1$; $P_1 = 0$), and $h(\mu) = (\alpha_0 + \alpha_2) + (\beta_0 + \beta_2) D$ for the gray zone of HP ($P_0 = 0$; $P_1 = 1$). The regression parameters $(\alpha_0, \alpha_1, \alpha_2, \beta_0, \beta_1, \beta_2)$ can be estimated by using the betareg package in R [21, 22]. To account for the day-specific variability in the outcome variable, the precision parameter was regressed by D with the log link.

Let $\mu(d)$ denote the true proportion of green area on $D = d$ days. The proportion of green area purely due to weed growth must be equal to zero on day $d = 0$, but the observed proportion of green area may be greater than zero due to background noise such as surrounding factors (e.g., light), human errors, and unknown reasons. To control the background noise, we define the adjusted proportion of green area on day $d$ as $\theta(d) = [\mu(d) - \mu(0)] / [1 - \mu(0)]$.

The rstanarm package was used to approximate the joint posterior distribution of regression parameters [23]. It was hypothesized that the weed growth cannot decrease with respect

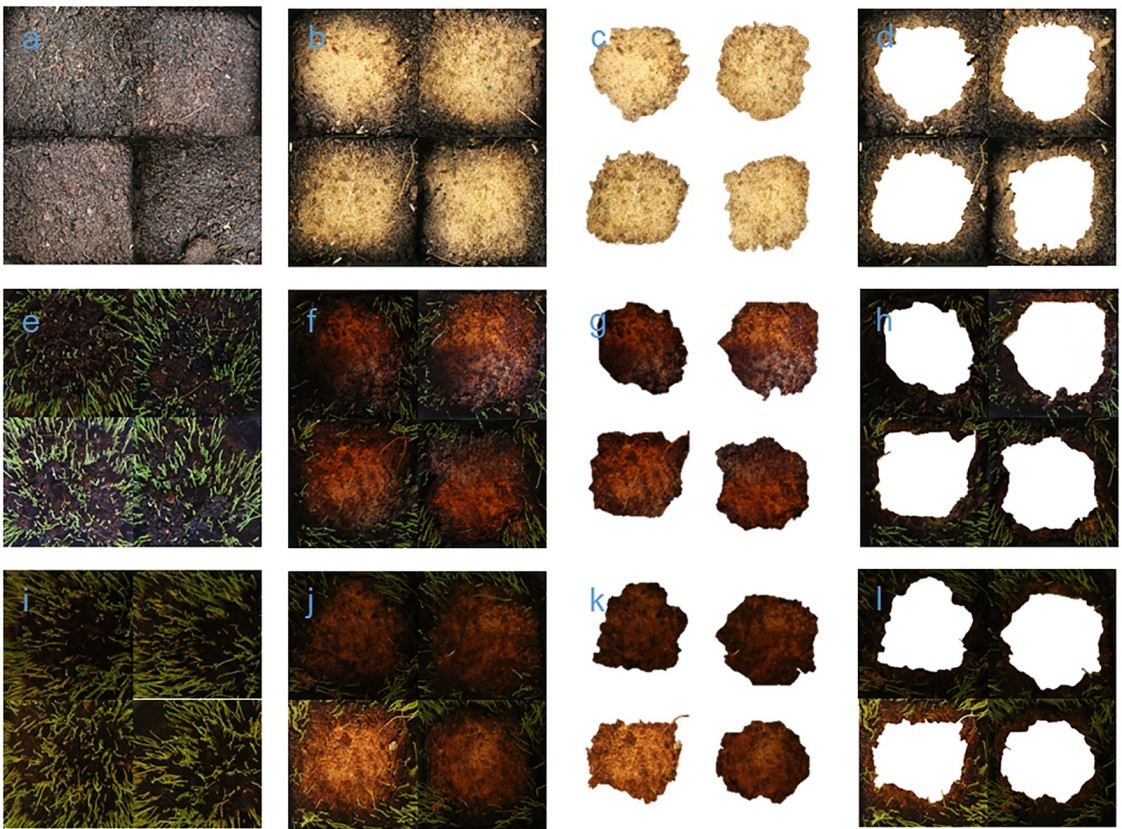

**Fig 2. The pictures of 2, 4, and 5 DAT.** a, b, c, and d: 2 DAT; e, f, g, and h: 4 DAT; i, j, k, and l: 5 DAT. a, e, and i: Control; b, f, and j: HP treatment (before segmentation); c, g, and k: Area directly affected by HP (P1; after segmentation); d, h, and l: Area not directly affected by HP (P0; after segmentation).

to time, so the parameter space was restricted as $\beta_0 > 0$, $\beta_0 + \beta_1 > 0$, and $\beta_0 + \beta_2 > 0$. The adjusted proportion on day $d$ at C, $P_0$, and $P_1$ is given by

$$\theta_C(d) = \frac{\mu_C(d) - \mu_C(0)}{1 - \mu_C(0)} = \frac{e^{\alpha_0 + \beta_0 d} - e^{\alpha_0}}{1 + e^{\alpha_0 + \beta_0 d}}, \tag{1}$$

$$\theta_0(d) = \frac{e^{\alpha_0 + \alpha_1 + (\beta_0 + \beta_1)d} - e^{\alpha_0 + \alpha_1}}{1 + e^{\alpha_0 + \alpha_1 + (\beta_0 + \beta_1)d}}, \tag{2}$$

and

$$\theta_1(d) = \frac{e^{\alpha_0 + \alpha_2 + (\beta_0 + \beta_2)d} - e^{\alpha_0 + \alpha_2}}{1 + e^{\alpha_0 + \alpha_2 + (\beta_0 + \beta_2)d}}, \tag{3}$$

respectively. The derivations of Eqs 1 to 3 are provided in S1 Appendix and the parameters of interest were $\theta_1(5)$ / $\theta_C(5)$ to compare $P_1$ to C on day 5 and $\theta_0(5)$ / $\theta_C(5)$ to compare $P_0$ to C on day 5. These parameters were estimated by approximate 95% credible intervals (CIs).

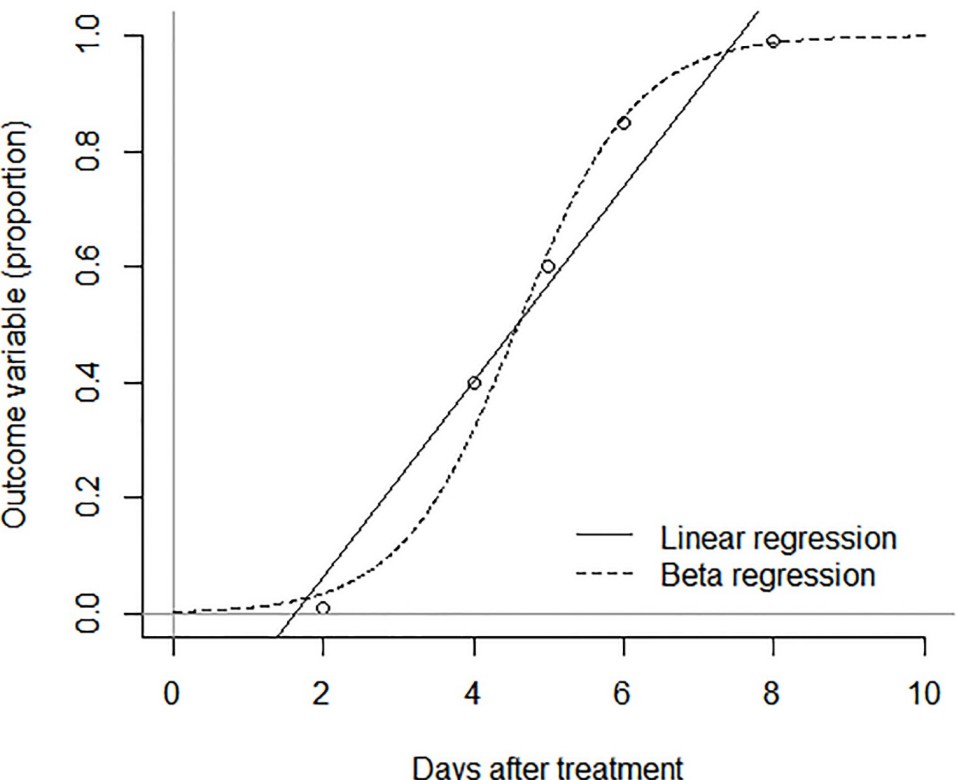

**Fig 3. Hypothetical data and estimated regressions to demonstrate a case when beta regression is more appropriate than linear regression.**

## Results

When HP was applied at a depth of 15 cm, bubbles were formed and covered the surface with heat. Subsequently the bubbled area turned gray (P1), but the other area (P0) did not show the same color (Figs 1 and 2). Noises can occur depending on imaging equipment and environment factors (e.g., light) during image acquisition. For the reason, we judged that the six most abundant green RGB codes (0-43-0, 51-85-51, 102-128-102, 153-170-102, 102-128-51, and 51-85-0), which appear in both control and treatment, fairly represent the actual weeds observed in the image. The other green RGB codes (i.e., G is the maximum among R, G, and B) are either noise or occupy very small pixels in the entire image (Fig 4).

The estimated regression parameters are graphically presented in Fig 5. The left panel shows the unadjusted proportion of green area (denoted by μ in the section of statistical analysis) estimated by the beta regression model before the image segmentation. The gray zone (P1) and the nongray zone (P0) were not separated in this analysis, so the HP treatment represents the combination of P0 and P1 which may not be an accurate representation of the treatment effect. In addition, due to the background noise, the estimated proportion of green area already exceeded 0.05 on day zero for the HP treatment and 0.1 in the control on day zero.

The right panel of Fig 5 shows the adjusted proportion of green area (denoted by $\theta_C$, $\theta_0$, and $\theta_1$ in Eqs 1, 2 and 3, respectively, in the section of statistical analysis) using the regression

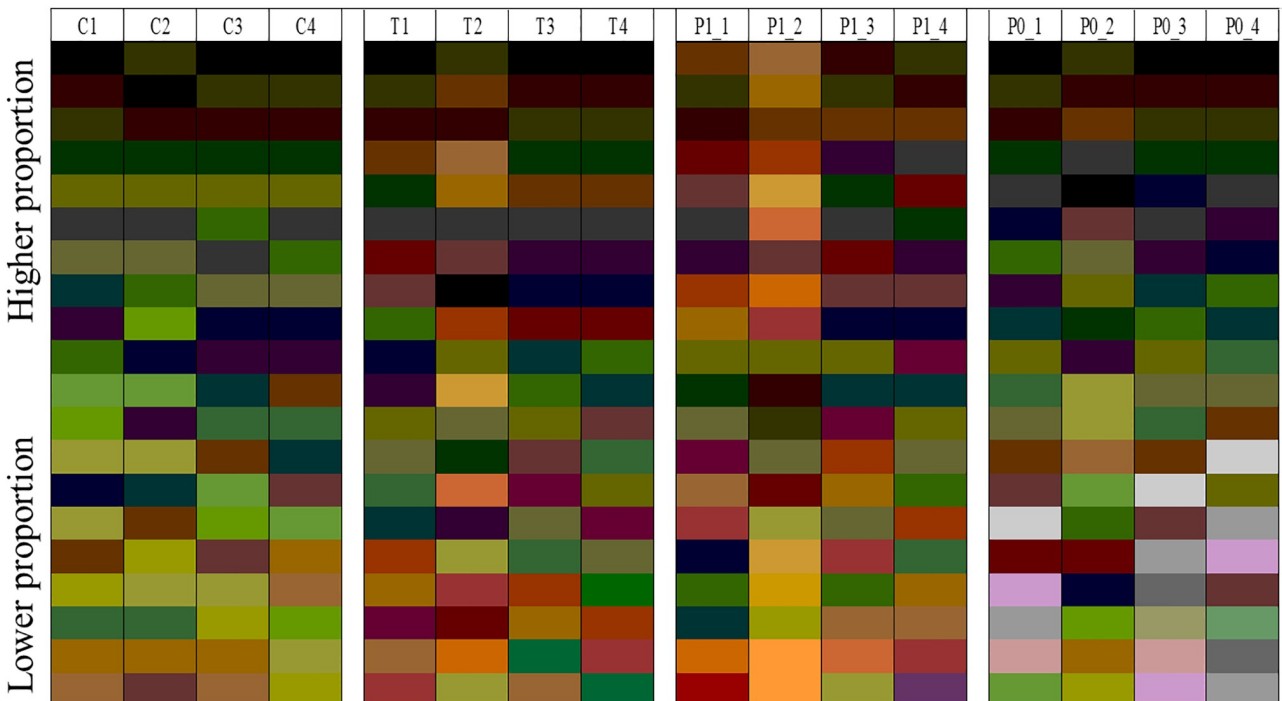

**Fig 4. The top 20 colors resulted by image analysis in pictures of 5 DAT.** C: Control; T: HP treatment (before segmentation); P1: Gray area directly affected by HP (after segmentation); P0: Nongray area not directly affected by HP (after segmentation).

parameters that are estimated under the beta regression after the image segmentation. In this analysis, P1 and P0 can be compared to C. For comparing P1 to C on day 5, the posterior mean of $\theta_1$ (5) / $\theta_C$ (5) was 0.12 with 95% CI of (0.0048, 0.37) which is entirely below one (i.e., P1 certainly has a low green proportion than C). In other words, we are 95% certain that the adjusted proportion of green area is between 0.0048 and 0.37 when we compare the gray zone of HP treatment to the control. For comparing P0 to C, the posterior mean of $\theta_0$ (5) / $\theta_C$ (5) was 1.12 with 95% CI of (0.50, 2.25) which includes one (i.e., high uncertainty to claim the difference between P0 and C). Based on the CIs, we conclude that the treatment effect is evident in P1, but not in P0. The approximate posterior distributions of $\theta_1$ (5) / $\theta_C$ (5) and $\theta_0$ (5) / $\theta_C$ (5) are shown in Fig 6.

## Discussion

Image segmentation is defined as follows: the search for homogeneous zones in an image and the classification of these zones [24]. Recently, image segmentation has been used widely including video analysis [25], medical image analysis [26], crack detection [27, 28], and plant disease recognition [29]. The application is extended to the pesticide efficacy assessment in this study. The image segmentation was applied for the two homogenous zones, gray and nongray, and the gray zone was produced because the hydroxyl radicals from HP induced decolorization [20]. The key idea of segmentation study is the distinguishability between the color of ground (e.g., soil) and the color of treated area. Therefore, this idea can be applied at the field-level experiments with other commercial pesticides. In addition, when pesticides have no color or their colors are not clearly distinguishable from the ground, pesticides can be mixed with dyes to indicate the effective areas of the pesticides. The coloration can be used for image

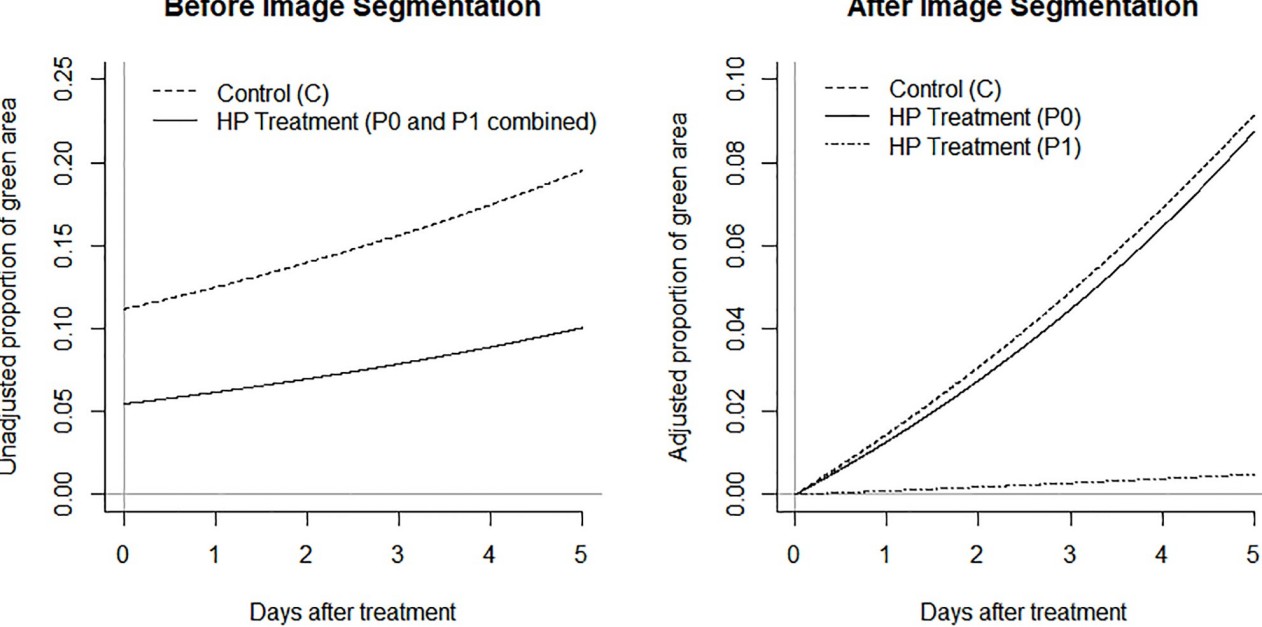

**Fig 5. The unadjusted proportion of green area before the image segmentation (left panel) and the adjusted proportion of green area by using the regression parameters estimated under the beta regression.**

segmentation immediately after using pesticides. In addition to the image segmentation, the adjusted proportion of green area was needed to accurately estimate the efficacy of HP relative to the control. Without the image segmentation and the adjustment, the relative unadjusted proportion of green area was 0.56 when HP was compared to the control on day 5. We believe that this is a substantially biased estimate. After the image segmentation and the adjustment, the relative proportion of green area was 0.12 on day 5. In fact, there was nearly no difference between the nongray zone of HP and the control, so the former analysis could have been a mixed result.

Image analysis has been used in other scientific areas such as medical informatics and technology, and the noise reduction in RGB data is an important step [30]. Despite the advance in quantitative image analysis software, observed RGB codes can vary among different softwares. In the context of agricultural data, noise in RGB data can occur by plant debris or minerals in agricultural fields, and this study demonstrates that the adjustment is meaningful and necessary to accurately quantify the treatment effect. Specifically, it is implausible to believe that about 10% of an experimental unit is occupied by green area as soon as the experiment began (the left panel of Fig 5). It would be more plausible to assume that the proportion of green area, which represents weeds, is close to zero at that time point (the right panel of Fig 5). In this study, the noise in RGB data is adjusted by statistical model rather than additional image processing. As such, the statistical adjustment method can be useful when a response variable (e.g., weed abundance) is identified by certain contrasting colors (e.g., green on brown/black background) with potential noise during image acquisition from the agricultural field.

HP reacts to produce hydroxyl radicals (Haber-Weiss reaction). The reaction can be catalyzed by transition metals such as iron and copper (Fenton reaction) [31]. HP efficiently initiates lipid peroxidation in polyunsaturated fatty acids of membrane [32]. HP, therefore, can destroy substances and cells by forming hydroxyl radicals. HP has been tested for control

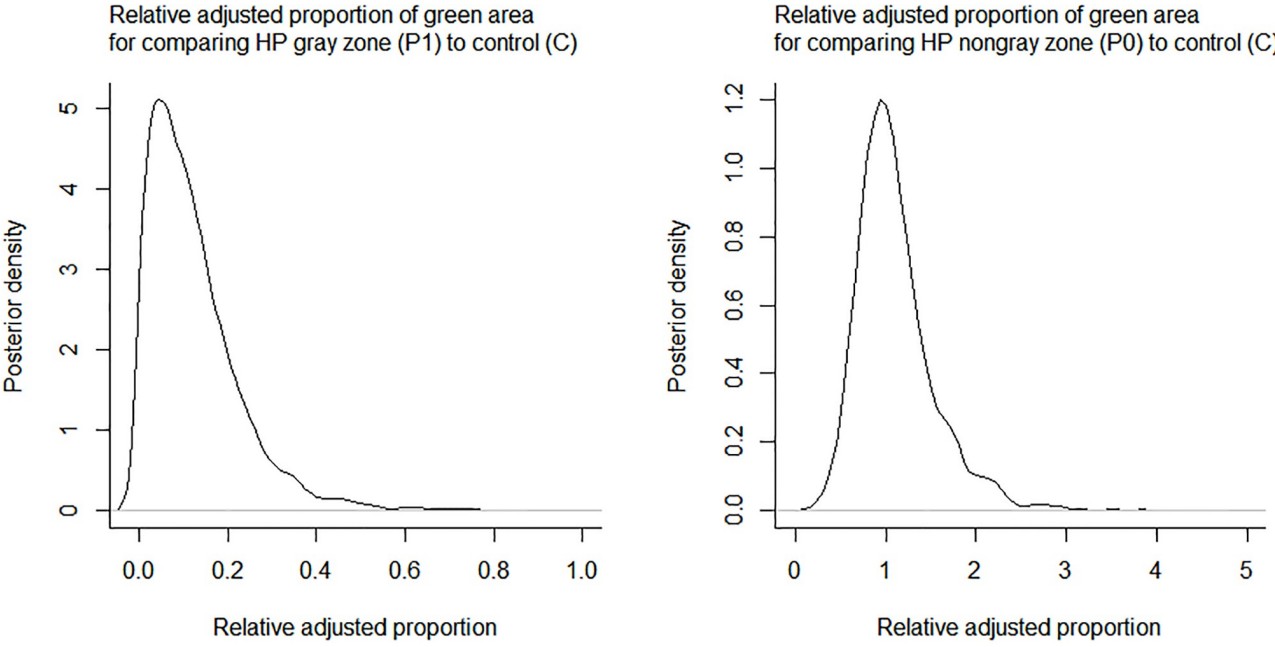

**Fig 6. The approximate posterior distributions of $\theta_1(5)/\theta_C(5)$ and $\theta_0(5)/\theta_C(5)$ (left panel and right panel, respectively).**

pathogens [33–35] and aquatic weeds [36, 37]. For this reason, we assumed that HP cannot only induce decolorization for the image segmentation but will also be effective in weed control for soil disinfestation although, as far as our literature review, there have not been any reports on the efficacy of HP for the terrestrial weed control. Additionally, the decomposition of HP liberates heat due to the degradation. The heat may reduce weed seed viability which is not affected by oxidative stress along with membrane damage due to the peroxidation of phospholipids by HP because of seed coat.

The cost-benefit should be considered for the pesticide efficacy assessment. Even though the efficacy of HP seemed clear at least for five days (or probably longer based on the estimated regression curves in Fig 5), the areas affected were limited (7.5 × 7.5 cm in this study). These results motivate us to reconsider the method of HP application that can maximize the benefit for a fixed cost. For instance, instead of injecting 12 mL of HP at the center, we could inject a higher concentration than 12 mL and/or spread the HP treatment to the edges of the flowerpots for a better weed control.

In most experiments, weeds grow eventually whether an experimental unit is treated or not, so the time of assessment can matter. To this end, we monitored experimental units over time and modeled the rate of weed growth. The rate of weed growth can be an important parameter for cost-benefit analysis. In this regard, the longitudinal analysis can bring more statistical information about the treatment effect than the cross-sectional analysis.

The combination of the image segmentation and longitudinal observation may be an economically and statistically efficient technique to accurately and precisely estimate the treatment effect. Therefore, it is possible to connect with various technologies for efficient weed removal. For instance, the combination can be used for real-time precision pesticide spraying systems [38, 39].

## Conclusion

We demonstrated the benefit of image segmentation and the necessity of adjustment in the proportion of green area. The image analysis can reduce the labor of data collection, and the image segmentation is helpful to accurately quantify the effect of HP relative to the control. However, the RGB analysis can be inaccurate due to the unexpected background noise in the RGB codes. To account for the nonzero background effect, the expected proportion can be adjusted after estimating the beta regression parameters. In addition, we suggest the longitudinal image segmentation instead of a cross-sectional assessment at an arbitrary time point. Finally, regarding the effect of HP, even though it seems to drastically reduce the germination of the ryegrass seeds, it can be applied in a different way or combined with other treatments such as solarization, steaming, and anaerobic soil disinfestation in the future studies.

## Supporting information

**S1 File. Colors, pixels, and RGB codes.** Colors, pixels, and RGB codes exported by the Image Area Analyzer (http://mkwak.org/imgarea/) when the picture from each flowerpot was uploaded in the website.
(XLSX)

**S1 Appendix.**
(DOCX)

## Author Contributions

**Conceptualization:** Steven B. Kim, Dong Sub Kim.

**Data curation:** Steven B. Kim, Dong Sub Kim, Xiaoming Mo.

**Formal analysis:** Steven B. Kim, Dong Sub Kim, Xiaoming Mo.

**Investigation:** Dong Sub Kim.

**Methodology:** Steven B. Kim.

**Resources:** Dong Sub Kim.

**Software:** Steven B. Kim.

**Supervision:** Steven B. Kim, Dong Sub Kim.

**Validation:** Steven B. Kim, Dong Sub Kim.

**Visualization:** Steven B. Kim, Dong Sub Kim, Xiaoming Mo.

**Writing – original draft:** Steven B. Kim, Dong Sub Kim, Xiaoming Mo.

**Writing – review & editing:** Steven B. Kim, Dong Sub Kim.

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
