## [Decision Letter · Decision Letter 0]

1 Feb 2021

PONE-D-20-33573

An image segmentation technique with statistical strategies for weed density assessment

PLOS ONE

Dear Dr. Kim,

Thank you for submitting your manuscript to PLOS ONE. After careful consideration, we feel that it has merit but does not fully meet PLOS ONE’s publication criteria as it currently stands. Therefore, we invite you to submit a revised version of the manuscript that addresses the points raised during the review process.

The manuscript requires a substantial amount of additional detail and explanation, which I think have been largely captured by the reviewers.  Therefore, address the points of each reviewer, including the two points listed by Reviewer #2.  What I think is needed is to ensure that a clear context for the method and application is provided, including actual field usage.  Reviewer #1 has captured many of the areas that need to be expanded and explained.

We look forward to receiving your revised manuscript.

Kind regards,

Randall P. Niedz

Academic Editor

PLOS ONE

Journal Requirements:

2) We suggest you thoroughly copyedit your manuscript for language usage, spelling, and grammar. If you do not know anyone who can help you do this, you may wish to consider employing a professional scientific editing service.  

3) PLOS requires an ORCID iD for the corresponding author in Editorial Manager on papers submitted after December 6th, 2016. Please ensure that you have an ORCID iD and that it is validated in Editorial Manager. To do this, go to ‘Update my Information’ (in the upper left-hand corner of the main menu), and click on the Fetch/Validate link next to the ORCID field. This will take you to the ORCID site and allow you to create a new iD or authenticate a pre-existing iD in Editorial Manager. Please see the following video for instructions on linking an ORCID iD to your Editorial Manager account: https://www.youtube.com/watch?v=_xcclfuvtxQ

Reviewers' comments:

Reviewer's Responses to Questions

**Comments to the Author**

1. Is the manuscript technically sound, and do the data support the conclusions?

Reviewer #1: Partly

Reviewer #2: Yes

2. Has the statistical analysis been performed appropriately and rigorously? 

Reviewer #1: Yes

Reviewer #2: Yes

3. Have the authors made all data underlying the findings in their manuscript fully available?

Reviewer #1: No

Reviewer #2: Yes

4. Is the manuscript presented in an intelligible fashion and written in standard English?

Reviewer #1: Yes

Reviewer #2: Yes

5. Review Comments to the Author

Reviewer #1: The manuscript deals with fusion of statistical method and image segmentation to quantify the weed density as a means to assess chemical treatment efficacy. The article addresses the common problem of biasness in weed assessment with image segmentation and offers a unique perspective to deal with such biasness. However, the manuscript lacks substantial literature review, clarity in the methodology, and important discussion to justify their work. I recommend major revision before it can be accepted for publication.

Introduction

Major:

• The authors use the term “weed density assessments” frequently throughout the manuscript but fail to imply the consistent meaning of this term throughout the study. Weed density assessment is a broader term and doesn’t only apply to assessing weeds that escape the herbicide/pesticide application. This term is more often used while assessing the weed infestation in a crop by measuring population parameters. The term “Herbicide/pesticide efficacy assessment” may be the better fit here.

• The authors have conducted the segmentation study at a very small experimental unit size. Moreover, their ability in performing a successful segmentation may root from what they used as chemical for the control. Because of such circumstances, it is very important that authors provide enough reasoning how this study can extrapolate to common practices of herbicide-based weed control at the field level. Also, how the relevancy of this technique holds for other chemicals (i.e. chemicals that may not show as distinct coloration as hydrogen peroxide) should be discussed.

• The introduction lacks substantial literature review. The problem authors pose in this study can be tackled by already established segmentation models (would say, just a simple color thresholding would work). Several approaches including, color space transformation and deep feature maps learning using deep neural networks have already proven to be effective for intricated segmentation task. However, the technique authors used in this study to adjust for biased estimate of green cover at early days of application may be unique and add new dimension to already existing pool of knowledge. Studies on ryegrass segmentation from crops have already been done in past. Below are recent papers on detecting ryegrass (the species author used for segmentation in this study) that authors should consider while developing literature review.

o https://www.mdpi.com/2072-4292/12/18/2977

o https://www.ncbi.nlm.nih.gov/pmc/articles/PMC6836412/

Minor:

• Line 50-54: What authors claim may need more reasoning. The fact that two or more distinctive zones appear within the experimental unit shouldn’t underestimate the scope of image analysis for accurate herbicide efficacy estimation. The first sentence should be cautiously rephrased.

• Line 72-74. Are authors trying focus on the herbicide efficacy on weed seed germination suppression or suppression of already germinated weed? This is unclear.

• Line 74-75: What is too late here? Why would the assessment be complex in this case? The chemical sprayed to suppress the weed seed germination are expected to work for longer time and it would be of interest to assess if they are vigorous for longer period. It doesn’t matter even if weed grows back and how complicated it would be; it should be evaluated to get an idea how effective the chemical is for long time. The author should provide more reasoning to this.

Materials and Methods:

Major:

• The core statistical technique that the authors used in this study has not been elaborated in detail. There is a possibility that the readers can be swayed away from the main theme of the technique.

• Several paragraphs are shorter and would be better to merge them wherever possible.

Minor:

• Treatment description and application and Data collection section can be merged if possible or each paragraph should be extended in the length.

• Line 108: The lighting conditions should be discussed.

• It is less common to explicitly mention the author of the software in such arrangement.

• Line 111: What is RGB code? It should be explicitly described.

• Was “Remove Background” process automatic or manual? Did the authors went through several rounds of trial to get the best segmentation? Please discuss the process as this process is crucial in this study.

• Line 130: What is Beta regression? Why it is used and why should it be used over linear regression? A reasoning that linear regression doesn’t impose bounds for an outcome variable doesn’t illuminate on how beta regression helps for this kind of problem.

• The equation numbers for each of the equation should be provided and referenced in the text.

• What is α0, α1, β0, and β1? What are predicted and predictor variables in the equation. They should be explicitly mentioned.

• The authors should explain in more detail how θ(d) is derived. Equation and parameter referencing is not enough here. θ(d) is the most important parameter in this study, so should be explicitly described.

• In fig. 1. If black color has been used in sub-plot (c) to denote the non-concerned area, it should be the same case with sub-plot (d). Why white color instead of black color in sub-plot (d)

Results

Major:

• The figures need to be revised for better appearance and veracity. The graphs need more elaboration. The results, however, looks promising and relate closely to the objectives.

Minor:

• Line 187: RGB code concept isn’t clear yet. How does finding the abundance of certain RGB color codes relate to estimating the outcome variable (i.e. adjusted proportion)?

• 189: Is it regression parameters that are estimated or the outcome variable as shown in figure 4?

• Figure 4 is not clear and line representation couldn’t be told apart.

• In figure 5, Y-axis is missing (seems like the header is the y-axis). It needs revision.

Discussion:

Major:

• Discussion seems to be oriented towards discussing the benefits of hydrogen peroxide and cost-benefits analysis. The authors should add a paragraph to heavily discuss what implications do finding the adjusted green portion hold for pesticide efficacy assessment (weed density assessment in authors word). Few examples of the scenarios where this adjustment technique could be meaningful at the field level should be provided.

Minor:

• Line 256- 258: Repetition of what’s already been discussed in intro section.

Conclusion:

• Conclusion is well written and conforms what have been found in the study.

Minor:

• Line 275-276: Longitudinal vs cross-section concept should be elaborated in the intro or method section before they could be introduced in the conclusion section.

Reviewer #2: The paper proposes to evaluate the efficacy of HP for weed control based on image segmentation and statistical modeling. Overall the essay is polished and easy to follow. However, there are two technical questions worthy of some further discussions.

Firstly, while calculating the portion of the green areas per image, why did the authors only choose the six most common RGB codes that meet the requirement of G > R and G > B? Any practical or theoretical support for this?

Secondly, one interesting result we can see from Fig. 5 is that the mean relative adjusted portion of the green area for P0 versus C is 1.12 and any number up to 2.25 is also within its 95% CI. Does it mean that the HP may promote the weed growth in some scenarios? At least in practice, it is hard to imagine such cases that the weed densities in P0 can exceed those in C.

In summary, the paper is well-written. And it would be great if the authors can add some more details about how did they choose the related parameters while implementing the proposed method and how to understand some of the deduced results.

6. PLOS authors have the option to publish the peer review history of their article (what does this mean?). If published, this will include your full peer review and any attached files.

Reviewer #1: No

Reviewer #2: No

---

## [Author Response · Author response to Decision Letter 0]

15 Feb 2021

Reviewer #1: The manuscript deals with fusion of statistical method and image segmentation to quantify the weed density as a means to assess chemical treatment efficacy. The article addresses the common problem of biasness in weed assessment with image segmentation and offers a unique perspective to deal with such biasness. However, the manuscript lacks substantial literature review, clarity in the methodology, and important discussion to justify their work. I recommend major revision before it can be accepted for publication.

Introduction

Major:

• The authors use the term “weed density assessments” frequently throughout the manuscript but fail to imply the consistent meaning of this term throughout the study. Weed density assessment is a broader term and doesn’t only apply to assessing weeds that escape the herbicide/pesticide application. This term is more often used while assessing the weed infestation in a crop by measuring population parameters. The term “Herbicide/pesticide efficacy assessment” may be the better fit here.

Response: Thank you for your comment and suggestion. We changed “weed density assessment” to “pesticide efficacy assessment” throughout the manuscript.

• The authors have conducted the segmentation study at a very small experimental unit size. Moreover, their ability in performing a successful segmentation may root from what they used as chemical for the control. Because of such circumstances, it is very important that authors provide enough reasoning how this study can extrapolate to common practices of herbicide-based weed control at the field level. Also, how the relevancy of this technique holds for other chemicals (i.e. chemicals that may not show as distinct coloration as hydrogen peroxide) should be discussed.

Response: Thank you for your comments. We inserted “Moreover, this idea can be applied at the field-level experiments with commercial pesticides. For example, when pesticides have no color or their colors are not clearly distinguishable from the ground, pesticides can be mixed with dyes to indicate the effective areas of the pesticides. Therefore, the coloration can be used for image segmentation immediately after using pesticides.” on line 253-256 in the Discussion section. 

• The introduction lacks substantial literature review. The problem authors pose in this study can be tackled by already established segmentation models (would say, just a simple color thresholding would work). Several approaches including, color space transformation and deep feature maps learning using deep neural networks have already proven to be effective for intricated segmentation task. However, the technique authors used in this study to adjust for biased estimate of green cover at early days of application may be unique and add new dimension to already existing pool of knowledge. Studies on ryegrass segmentation from crops have already been done in past. Below are recent papers on detecting ryegrass (the species author used for segmentation in this study) that authors should consider while developing literature review.

o https://www.mdpi.com/2072-4292/12/18/2977

o https://www.ncbi.nlm.nih.gov/pmc/articles/PMC6836412/

Response: Thank you for your comments and recommendations. We inserted “Ryegrass (Lolium multiflorum) is one of the most predominant weeds in United States [1] and pesticides are used to control ryegrass” on line 41-42 and “Several approaches using deep neural networks have proven to be effective for segmentation task [12] [13] [14]. Moreover, detection of ryegrasses in crops using machine learning technology [15] and detection of other weeds in ryegrasses using deep learning technology [16] have been reported” on line 64-67.

Minor:

• Line 50-54: What authors claim may need more reasoning. The fact that two or more distinctive zones appear within the experimental unit shouldn’t underestimate the scope of image analysis for accurate herbicide efficacy estimation. The first sentence should be cautiously rephrased.

Response: Thank you for your comment. We deleted “Sometimes, the efficacy of a pesticide cannot be proven by image analysis nor hand counting because the pesticide may not affect an entire experimental unit” and inserted “If the pesticide does not affect an entire experimental unit, and if the researcher assesses the efficacy on the entire experimental unit, the measure of efficacy can be inaccurate and/or unstable. In this regard, the image analysis will be more useful than manual counting.” on line 52-54. 

• Line 72-74. Are authors trying focus on the herbicide efficacy on weed seed germination suppression or suppression of already germinated weed? This is unclear.

Response: Thank you for your comment. We are trying focus on weed seed germination suppression. So, we inserted “on suppression of weed seed germination” on line 79-80.

• Line 74-75: What is too late here? Why would the assessment be complex in this case? The chemical sprayed to suppress the weed seed germination are expected to work for longer time and it would be of interest to assess if they are vigorous for longer period. It doesn’t matter even if weed grows back and how complicated it would be; it should be evaluated to get an idea how effective the chemical is for long time. The author should provide more reasoning to this.

Response: Probably, our language “too late” was not clear. Our main point was that the effect size relative to the control may be a function of time (because the treatment effect does not last forever), and it would be more reasonable to observe data longitudinally rather than a cross-sectional observation (line 79-88). Your comment is appreciated.

Materials and Methods:

Major:

• The core statistical technique that the authors used in this study has not been elaborated in detail. There is a possibility that the readers can be swayed away from the main theme of the technique.

Response: We agree that the statistical model could be elaborated in detail. We expressed the model with more detailed explanations with an added Fig 3 to help readers (line 140-156). Thank you for your suggestion.

• Several paragraphs are shorter and would be better to merge them wherever possible.

Response: Thank you for your suggestion. We merged the Treatment description and application and Data collection section.

Minor:

• Treatment description and application and Data collection section can be merged if possible or each paragraph should be extended in the length.

Response: Thank you for your suggestion. We merged the Treatment description and application and Data collection section.

• Line 108: The lighting conditions should be discussed.

Response: Thank you for your comments. We inserted “under the lights of four incandescent bulbs of 20 watts” on line 117.

• It is less common to explicitly mention the author of the software in such arrangement.

Response: Agreed. We removed it.

• Line 111: What is RGB code? It should be explicitly described.

Response: Thank you for your comments. We inserted “which represent various colors quantitatively as a combination of red, green and blue” on line 119-120.

• Was “Remove Background” process automatic or manual? Did the authors went through several rounds of trial to get the best segmentation? Please discuss the process as this process is crucial in this study.

Response: Thank you for your comments. We inserted “The process is automatic and has been tried several times to obtain the best segmentation” on line 128-129.

• Line 130: What is Beta regression? Why it is used and why should it be used over linear regression? A reasoning that linear regression doesn’t impose bounds for an outcome variable doesn’t illuminate on how beta regression helps for this kind of problem.

Response: Beta regression is a special kind of regression which can model data especially when their values are between zero and one (Ferrari and Cribari-Neto, 2004). If linear regression is used for our data (the outcome variable ranges between zero and one), it results in biased estimation. To make this point clear, we revised the fifth paragraph of the introduction (line 89-102), and we added a figure of hypothetical data which illustrates the caveat of using linear regression (Fig 3). Your comment is greatly appreciated, and we believe that the revision is helpful to illuminate the point.

• The equation numbers for each of the equation should be provided and referenced in the text.

Response: The three equations are numbered (1) to (3) (line 167-178), and they are referred in the results section (line 216-218).

• What is α0, α1, β0, and β1? What are predicted and predictor variables in the equation. They should be explicitly mentioned.

Response: (α0, α1, α2, β0, β1, β2) are regression parameters to be estimated given data. In the beta regression, in our study. The observed proportion of green area per image is the outcome variable (to be predicted). The days after treatment and treatment group (C, P0, or P1) are the predictor variables. In the revised manuscript, when we elaborated in the section of statistical analysis (to address your previous question), we explicitly mentioned the parameters, predicted variable, and predictor variables (line 140-151). Thank you for your suggestion.

• The authors should explain in more detail how θ(d) is derived. Equation and parameter referencing is not enough here. θ(d) is the most important parameter in this study, so should be explicitly described.

Response: Thank you for the suggestion. We decided that using the Appendix is appropriate for the detailed derivations, and we directed readers to the Appendix on line 320.

• In fig. 1. If black color has been used in sub-plot (c) to denote the non-concerned area, it should be the same case with sub-plot (d). Why white color instead of black color in sub-plot (d)

Response: Thank you for your comment. We changed black background to white.

Results

Major:

• The figures need to be revised for better appearance and veracity. The graphs need more elaboration. The results, however, looks promising and relate closely to the objectives.

Response: To improve the quality of Fig 4 and Fig 5, we will submit the figures in PDF files as well. We added labels of the x-axis and y-axis of Fig 5, and we adjusted the y-axis of Fig 5 to clearly distinguish two estimated regression curves (as you suggested in your minor comment). Thank you for your comments.

Minor:

• Line 187: RGB code concept isn’t clear yet. How does finding the abundance of certain RGB color codes relate to estimating the outcome variable (i.e. adjusted proportion)?

Response: Thank you for your comment. We have seen some measurement errors in the RGB codes in previous studies (also discussed in the discussion section). In particular, noises can occur depending on imaging equipment and environmental factors (e.g., light) during image acquisition. For the reason, we judged that the six most abundant green RGB codes (0-43-0, 51-85-51, 102-128-102, 153-170-102, 102-128-51, and 51-85-0), which appear in both control and treatment, fairly represent the actual weeds observed in the image. The other green GRB codes (i.e., G is the maximum among R, G, and B) are either noise or occupy very small pixels in the entire image. This explanation is added in the first paragraph of the result section (line 201-206). 

• 189: Is it regression parameters that are estimated or the outcome variable as shown in figure 4?

Response: It is the regression parameters that are estimated. The caption of Fig 5 and line 237-239 are revised to make this point clearer.

• Figure 4 is not clear and line representation couldn’t be told apart.

Response: The right figure is magnified by showing the y-axis from 0 to 0.1 so that the lines for C and P0 are better distinguishable.

• In figure 5, Y-axis is missing (seems like the header is the y-axis). It needs revision.

Response: We added the y-axis, and we labeled both x-axis (relative adjusted proportion) and y-axis (posterior density) to clearly present the results. Thank you.

Discussion:

Major:

• Discussion seems to be oriented towards discussing the benefits of hydrogen peroxide and cost-benefits analysis. The authors should add a paragraph to heavily discuss what implications do finding the adjusted green portion hold for pesticide efficacy assessment (weed density assessment in authors word). Few examples of the scenarios where this adjustment technique could be meaningful at the field level should be provided.

Response: Thank you for your comment. We added a paragraph to address your points (the third paragraph of the discussion section in the revised manuscript; line 264-276). Thank you for your suggestions. 

Minor:

• Line 256- 258: Repetition of what’s already been discussed in intro section.

Response: Thank you for your comment. We deleted the repetition.

Conclusion:

• Conclusion is well written and conforms what have been found in the study.

Minor:

• Line 275-276: Longitudinal vs cross-section concept should be elaborated in the intro or method section before they could be introduced in the conclusion section.

Response: We elaborated longitudinal vs. cross-section concept in the introduction section (lines 79-88).

 

Reviewer #2: The paper proposes to evaluate the efficacy of HP for weed control based on image segmentation and statistical modeling. Overall the essay is polished and easy to follow. However, there are two technical questions worthy of some further discussions.

Firstly, while calculating the portion of the green areas per image, why did the authors only choose the six most common RGB codes that meet the requirement of G > R and G > B? Any practical or theoretical support for this?

Response: Thank you for the suggestion. Your comment is very similar to a comment of another reviewer, so both comments are addressed simultaneously in the first paragraph of the result section (line 201-206). 

Secondly, one interesting result we can see from Fig. 5 is that the mean relative adjusted portion of the green area for P0 versus C is 1.12 and any number up to 2.25 is also within its 95% CI. Does it mean that the HP may promote the weed growth in some scenarios? At least in practice, it is hard to imagine such cases that the weed densities in P0 can exceed those in C.

Response: (In the revised manuscript, Fig 5 is now Fig 6.) We agree that it is hard to imagine the promotion of weed growth by HP. The long range of the 95% CI (from 0.5 to 2.25) is due to the uncertainty in our sample. The key point is that the mean is close to 1 and the 95% CI captured 1 when we compare P0 to C, whereas the 95% CI for comparing P1 to C is entirely far below 1 (statistically significant evidence for the treatment effect). This point is clearly made in the revised manuscript (line 216-225), and it will be helpful for the readers. Your comment is greatly appreciated.

In summary, the paper is well-written. And it would be great if the authors can add some more details about how did they choose the related parameters while implementing the proposed method and how to understand some of the deduced results.

---

## [Editor Report · Decision Letter 1]

2 Mar 2021

An image segmentation technique with statistical strategies for pesticide efficacy assessment

PONE-D-20-33573R1

Dear Dr. Kim,

We’re pleased to inform you that your manuscript has been judged scientifically suitable for publication and will be formally accepted for publication once it meets all outstanding technical requirements.

Kind regards,

Randall P. Niedz

Academic Editor

PLOS ONE
---

## [Editor Report · Acceptance letter]

4 Mar 2021

PONE-D-20-33573R1 

An image segmentation technique with statistical strategies for pesticide efficacy assessment 

Dear Dr. Kim:

I'm pleased to inform you that your manuscript has been deemed suitable for publication in PLOS ONE. Congratulations! Your manuscript is now with our production department. 

Kind regards, 

on behalf of

Dr. Randall P. Niedz 

Academic Editor

PLOS ONE